# Production of Cationic Starch-Based Flocculants and Their Application in Thickening and Dewatering of the Municipal Sewage Sludge

**DOI:** 10.3390/ma16072621

**Published:** 2023-03-25

**Authors:** Edita Lekniute-Kyzike, Joana Bendoraitiene, Vesta Navikaite-Snipaitiene, Laura Peciulyte, Ramune Rutkaite

**Affiliations:** Department of Polymer Chemistry and Technology, Kaunas University of Technology, Radvilenu Rd. 19, 50254 Kaunas, Lithuania

**Keywords:** modified starch, flocculant, municipal sewage sludge

## Abstract

Polymer flocculants are used to promote solid–liquid separation processes in wastewater treatment technologies, and bio-based flocculants possess many advantages over conventional synthetic polymers. Potato starch microgranules were chemically modified and mechanically sheared to produce modified starch flocculants. The effectiveness of produced cationic starch (CS) and cross-linked cationic starch (CCS) flocculants in the thickening and dewatering of surplus activated sewage sludge was evaluated and compared with that of synthetic cationic flocculants (SCFs) The flocculation efficiency of SCF, CS, and CCS in sludge thickening was determined by measuring the filtration rate of treated surplus activated sludge. Comparing the optimal dose of SCFs and CCS flocculants needed for thickening, the CCS dose was more than 10 times higher, but a wide flocculation window was determined. The impact of used flocculants on the dewatering performance of surplus activated sludge at optimal dose conditions was investigated by measuring capillary suction time. The filtration efficiencies (dewaterability) of surplus activated sludge using SCF, CS, and CCS were 69, 67, and 72%, respectively. The study results imply that mechanically processed cross-linked cationic starch has a great potential to be used as an alternative green flocculant in surplus activated sludge thickening and dewatering operations in municipal sewage sludge treatment processes.

## 1. Introduction

One of the biggest environmental problems today is urban wastewater recycling and its utilization. With the acceleration of urbanization, human activity, and global industrialization, the amount of generated municipal sewage sludge is rapidly increasing [1]. Sludge is formed in all wastewater treatment operations. Sludge of various types such as primary raw sludge, surplus activated sludge, thickened activated sludge, mixed sludge, digested sludge, etc., are generated at different treatment stages [2]. The cost of sludge treatment and disposal operations is strongly affected by its volume, water content, or solids concentration. The sludge treatment and disposal may account for up to 60% of the total operation cost in the wastewater treatment plant [3]. Hence, thickening and dewatering are important steps in the total sludge processing.

Flocculation is a widely used process in many fields of industry, including wastewater or sludge treatment. Various flocculants and coagulants including inorganic, organic synthetic and natural flocculants are used for sludge thickening and dewatering [4]. Most synthetic flocculants are derived from petroleum products and are toxic to humans, animals, and aquatic organisms [5,6,7]. For example, acrylamide, which has residues that may be present in polyacrylamide-based flocculants, is classified as an extremely hazardous substance and is included in the list of carcinogenic substances [8]. Hence, it is important to find biobased alternatives to synthetic flocculants that would have the same or similar efficiency as synthetic ones, would be obtained from renewable resources, and would be relatively harmless to the environment.

Recent studies focus on the use of readily available, safe, and cheap biopolymers, particularly polysaccharides such as starch, cellulose, and chitosan [9]. To improve their flocculation efficiency, polysaccharides are subjected to chemical modification. One of the natural polymers used for the preparation of such flocculants is starch, which is biodegradable and has environmentally friendly decomposition products. In general, starch cannot directly act as a flocculant and should be modified [10]. Mostly, cationic functional groups are introduced onto the starch backbone [11]. Cationic starches containing quaternary ammonium groups are effective over a wide range of pHs. Moreover, native starch can be cross-linked with epichlorohydrin in the presence of alkali [12]. Cross-linking is another important factor when imparting functional properties to starch as it provides a lower viscosity of paste, higher firmness of the granules, higher thermal stability, etc. Thus, cationic starch derivatives could be a suitable alternative to inorganic and synthetic organic flocculants in wastewater treatment. However, there are no data about the testing cross-linked cationic starches as flocculants so far.

Starch-based flocculants have been applied for the dewatering of the surplus sludge by flocculating the sludge particles [10,13,14]. The sludge dewatering performance could be evaluated by assessing the filtration time, capillary suction time, specific resistance to filtration and sludge volume after sedimentation, etc. The research results [13] indicated that the optimum dosage of combined aluminum-ferrous-starch flocculant for sludge dewatering was 8.8 g/kg dry sludge, but the process depended on the pH of the medium. A water-soluble copolymer—starch-graft-poly (2-methacryloyloxyethyl) trimethyl ammonium chloride—was used as a dewatering agent for anaerobic sludge, and it was determined that sludge could easily be filtered using the dosage of 0.7% of the dry weight of sludge [10]. In another study, the sludge dewatering agent was synthesized by grafting copolymerization of enzymatically treated starch and dimethyldiallylammonium chloride (DMDAAC) using horseradish peroxidase/H_2_O_2_ initiation [14]. The specific resistance to filtration and capillary suction time of the sludge using starch-g-p(DMDAAC) decreased, and the sludge water content could be reduced from 98% to 51%. In another paper [15], the graft copolymers of acrylamide and allyl trimethyl ammonium chloride using soluble starch were prepared. The obtained copolymers were suggested as new flocculants for wastewater treatment due to their low dose and effectiveness in a wide temperature and pH range.

The comparative data of various coagulants and flocculants used in sludge dewatering is summarized in Table 1 [4].

Inorganic coagulants and synthetic polymeric flocculants are the most common agents used in the dewatering process [16,17]. Inorganic coagulants such as polyaluminium chloride and ferric chloride are low cost and have a moderate effect on dewatering. However, they also have many disadvantages, such as high dose, sensitivity to pH and share force, and biological toxicity caused by residual metal ions [4]. Synthetic polymeric flocculants including cationic polyacrylamide or combined materials are excellent in terms of dose and dewatering efficiency [18], but their cost is relatively high. However, such types of flocculants may cause secondary pollution and health hazards due to remaining unreacted monomers [4]. Consequently, synthetic polymeric flocculants that are common in the current market and lead to secondary pollution to the environment should be, if possible, replaced by natural polymeric flocculants.

Natural polymeric flocculants such as starch and chitosan have many advantages, such as widespread availability, biodegradability, and low toxicity [19,20,21]. Furthermore, these types of flocculants act in a wide range of pHs with good dewatering efficiency using a relatively low dose of material [4]. In addition, natural polymeric flocculants can be combined with inorganic coagulants or synthetic polymers such as ferric chloride [22], polyaluminium chloride [23], or polyacrylamide [24,25] to improve the dewatering performance.

There are cationic starch products, such as Greenfloc 120 (Hungary), C*Bond HR (England), and Emfloc KC 750 (Germany), available on the market. However, Greenfloc 120 is sold as a solution that is limiting its application areas and is economically disadvantageous. The degree of cationic groups substitution in C*Bond HR is low, so the flocculation efficiency and cold-water solubility are also low. In addition, using flocculants such as Emfloc KC 750 with a high degree of substitution (DS > 0.54), the biodegradability of the product can be reduced [26]. Hence, to maintain the biodegradability of polysaccharide, the starch-based flocculants with lower DS should be prepared.

The aim of the work was to investigate the suitability of cationic starch-based flocculants for municipal sewage sludge thickening and dewatering, and to compare their effectiveness with the widely used synthetic cationic flocculant. For the preparation of the cationic starch and cross-linked cationic starch flocculants, potato starch was chemically modified and then mechanically processed using shear forces. Compared to other studies, a low amount of cationic reagent for modification of starch was used, but efficient flocculants were still obtained. An important advantage of the developed biobased polymer flocculants compared to existing industrial solutions is that the products were obtained as solid materials. The efficiency of the modified starches and synthetic flocculants in the thickening and dewatering of surplus activated sludge was determined by measuring the filtration flow and capillary suction times. Additionally, the thickening of the surplus activated sludge by using a dual flocculant system was explored. Summarizing the obtained results, the most promising flocculant for the thickening and dewatering of the municipal sewage sludge was recommended.

## 2. Materials and Methods

### 2.1. Materials

The native potato starch (intrinsic viscosity [η] = 0.39 L/g, MW = 10^3^–10^4^ kDa), containing 21–23 wt% of amylose and 77–79 wt% of amylopectin was received from Aloja-Starkelsen (Ungurpils, Latvia). Glycidyltrimethylammonium chloride (GTAC, 70%), epichlorohydrin (EPI, 98%), toluidine blue O, and kaolin (average particle size 0.9 μm; ζ-potential −31 mV) were supplied by Sigma–Aldrich. The dextran sulphate (MW = 500,000) was obtained from the Loba Feinchemie (Fischamend, Austria). The synthetic cationic flocculant (Praestol 859) was supplied by Ashland Inc. (Wilmington, DE, USA). The synthetic anionic flocculant (Fennopol A305) was obtained from Kemira (Helsinki, Finland). All other used chemicals were of analytical grade. The surplus activated sludge was obtained from municipal sewage treatment plant AB “Kauno Vandenys” (Kaunas, Lithuania), after aeration and settling in the tank operations. The important parameters of the activated sludge were as follows: pH 7.3; zeta potential, +3.53 mV; CST, 32.2 s; water content, 99.08%; and total solids, 0.92%.

### 2.2. The Synthesis of Cationic Starch Derivatives

The cationic starch (CS) was prepared by etherification of the native potato starch with glycidyltrimethylammonium chloride (GTAC) in the presence of sodium hydroxide as a catalyst at 45 °C for 48 h using the method described in our previous studies [11]. The molecular mass of the anhydroglucoside unit (AGU) was assumed as a mole of starch. The molar ratio of AGU:GTAC:NaOH:H_2_O was 1:0.2:0.04:3.5.

The cross-linked cationic starch (CCS) was prepared using a two-stage method [27]. Firstly, the starch was cross-linked with epichlorohydrin (EPI) in the presence of NaOH (the molar ratio AGU:EPI:NaOH:H_2_O was 1:0.005:0.06:10) at 45 °C for 24 h. In the second step, the cross-linked starch was modified using GTAC in the presence of sodium hydroxide as a catalyst at 45 °C for 48 h. The molar ratio of AGU:GTAC:NaOH:H_2_O was 1:0.23:0.04:3.

The nitrogen content of CS and CCS was estimated using the Kjeldahl method after purification using the Soxhlet extraction with methanol for 16 h. The degree of the substitution of cationic groups was calculated from the nitrogen content:(1)$$DS=\frac{162\cdot N}{1400-151.5\cdot N}$$
where *N* is the nitrogen content estimated using the Kjeldahl method (%), 162 is the molecular weight of the AGU, 1400 is the molecular weight of the nitrogen multiplied by a hundred, and 151.5 is the molecular weight of the (GTAC).

The degree of substitution of the quaternary ammonium groups in both CS and CCS was found to be 0.19.

### 2.3. Preparation of Cationic Starch Flocculants

The microgranules of cationic or cross-linked cationic starches were swollen to the equilibrium state in distilled water (1% *w*/*w*) and then processed using mechanical shearing at 15,000 rpm and different durations using an Ultra-Turrax T25 digital (IKA, Staufen, Germany) device at room temperature to obtain a flocculant colloidal dispersion. The prepared dispersions were used in all flocculation experiments.

### 2.4. Determination of Cationic Group’s Accessibility

To determine the accessibility (%) of the cationic groups of starch derivatives to polyanions, the polyelectrolyte titration was performed as described in our previous studies [28,29].

### 2.5. Determination of Flocculation Efficiency Using Model Kaolin System

The flocculation experiment with a model kaolin suspension was performed and the residual turbidity (RT) of the kaolin suspension after the addition of the CCS flocculant was evaluated according to the previously described method [30]. The flocculation efficiency of modified starch flocculants was characterized by the minimum amount (dose) of the flocculant C (mg per 1 g of kaolin) in the presence of which the kaolin suspension destabilization occurs up to 10% of RT and by the width of the flocculation window (W). W is defined as a difference between the maximum and minimum amounts of flocculant, at the presence of which RT is less than 10%.

### 2.6. Determination of Flocculation Efficiency Using Surplus Activated Sludge

The flocculation efficiency using SCF, CS, and CCS for sludge thickening and dewatering was determined by measuring the filtration rate and capillary suction time (CST) of the surplus activated sludge, respectively.

For the determination of the flocculation efficiency according to the filtration rate, 400 mL of the municipal sewage sludge suspension was poured into a 600 mL volume beaker, followed by dosing the required amount of the flocculating agent and the resulting disperse system mixed. For the dual flocculation system, 15 mg of synthetic anionic flocculant was added to 3 L of sludge before flocculation with the cationic flocculant. The experiment was carried out at room temperature. After the destabilization, 200 mL of sludge dispersion was filtered through the polyamide filter and the filtrate flow time—the filtration rate (*FE_R_*), mL/s—was calculated:(2)$${FE}_{R}=\frac{200}{t}$$
where 200—volume of filtrate (mL); and *t*—filtrate flow time (s).

For the dual flocculation system, the light absorption of the destabilized sewage sludge filtrate was also measured at the wavelength of 500 nm using the Unicam UV3 (Cambridge, England) UV/Vis spectrophotometer.

For the determination of the flocculation efficiency according to the capillary suction time, the dewatering of the surplus activated sludge was evaluated by using standard 304M CST equipment (Triton Electronics Ltd., Dunmow, UK), which measures the capillary suction time (CST). The capillary suction pressure generated by standard filter paper (Whatman Grade 17 Chr Cellulose Chromatography Paper) was used to ‘suck’ the water from the sludge. The filtration efficiency (*FE_T_*), %, was calculated according following formula:(3)$${FE}_{T}=\frac{t_{k}-t_{a}}{t_{k}}$$
where *t_k_*—dewatering time of dispersion system without flocculant, control sample (s); and *t_a_*—dewatering time of destabilized dispersion system (s).

Additionally, the filtration efficiency was assessed after the mixing of the thickened sludge slurries in a glass beaker with magnetic stirrer at 100, 500, and 1200 rpm stirring rate for 5 min.

### 2.7. The Optical Microscopy Studies

The optical observations of the CCS microgranules were carried out using an Olympus CX31 optical microscope (Cebu, Philippines) under 100-time magnification. The samples were prepared by dyeing cationic starch derivative microgranules with an 0.005 M standard iodine solution. The photographs of the CCS slurry were taken using an Olympus camera.

## 3. Results and Discussion

### 3.1. Preparation of Cationic Starch Based Flocculants

The cationic starch (CS) and cross-linked cationic starch (CCS) were prepared according to the methods described in our previous studies [11,31]. The CS with DS = 0.19 of the quaternary ammonium groups was obtained by performing the etherification reaction between the hydroxyl groups of the potato starch and glycidyltrimethylammonium chloride (GTAC) in the presence of sodium hydroxide as a catalyst. The introduction of quaternary ammonium groups into starch molecules was demonstrated using ^1^H NMR spectroscopy as previously described [11]. The CCS was obtained by performing a two-stage reaction (see Figure 1).

Firstly, the native potato starch was cross-linked with a low amount of epichlorohydrin (0.0005 mol/mol AGU) in an aqueous alkali medium. Afterward, the CCS with 0.19 DS of quaternary ammonium groups was obtained by performing the etherification reaction between hydroxyl groups of starch and GTAC in the presence of NaOH as a catalyst. The introduction of cationic groups and the appearance of the cross-linked structure solid-state have been proved by recording the ^13^C CP/MAS NMR spectrum as described in our previous study [31]. It can be emphasized that the starch is a cheap and rapidly renewable bio-based raw material that can be easily applied for the preparation of starch-based flocculants. The modification agents, such as epichlorohydrin and glycidyltrimethylammonium chloride, are widely used and low-cost reagents. In our study, the synthesis of CCS is simple with a high yield reaching up to 100%.

One of the main challenges when preparing effective flocculants is to maintain a high affinity for water and a sufficient availability of cationic groups of modified starches. Therefore, a small amount of cross-linking agent was used to obtain the water-insoluble CCS granules, but with high degree of swelling in the aqueous medium, as these properties were required for the further mechanical treatment of the CCS granules. When preparing cationic starch-based flocculants, the CS and CCS were swollen in water and then processed by using mechanical shearing; as a result, colloidal dispersions composed of ruptured and broken CS or CCS microgranules were obtained. The optical microscopy images revealed (see Figure 2) that the swollen CCS microgranules were crushed into submicron-sized particles during the mechanical processing.

The results of our previous study revealed that, after mechanical treatment, the CS derivatives had lower molecular weight, smaller particle size, higher accessibility to polyanions, and higher flocculation efficiency [11]. In this study, it was determined that, in the case of cross-linked cationic starch (Figure 3a), a longer shearing duration was required to achieve high cationic group’s accessibility values compared to those of CS.

As in the case of CS, the CCS accessibility to polyanions increased with the increasing shearing duration. The accessibility of CCS cationic groups was only 4%, and, after 30 min of shearing, it reached 40%. Meanwhile, in the case of CS, the accessibility of the cationic centers in the CS slurry was 18% and reached 90% after 30 min of shearing [11].

Based on the flocculation efficiency results (see Figure 3b), the optimal shearing duration for CCS slurry preparation was chosen. The flocculation efficiency of CCS using a model kaolin system was investigated using various duration of shearing and expressed as the minimum amount (dose) of the flocculant (C, mg/g) and the width of the flocculation window (W, mg/g). As can be seen from the results, after 2 min of shearing, the flocculation results were changed, i.e., C decreased from 137 to 34 mg/g and W changed from 363 to 155 mg/g. When the duration of shearing was increased to 30 min, C and W were decreased by 10 and 7.7 times from the initial value, respectively. These results correlate well with the data presented in Figure 3a, as the accessibility of the cationic groups was increased meant a lower amount of flocculant was required. The lower W value indicates the lower flocculant efficiency, but the low C value is desirable at the same time. Based on the results of the flocculation studies, 20 min of shearing duration of CCS was chosen for the preparation of flocculant dispersion. In the case of CS, an optimal shearing duration of 15 min was selected according to the previously obtained results [11].

### 3.2. Thickening of the Surplus Activated Sludge by Using Cationic Starch Based Flocculants

The suitability of CS and CCS as cationic starch flocculants for municipal sewage sludge thickening and their effectiveness compared with the widely used synthetic cationic flocculant (SCF) were investigated. The flocculation efficiencies using SCF, CS, and CCS were determined by measuring the filtrate flow time of the thickened surplus activated sludge (see Figure 4).

As can be seen from the results, the CS was not effective as the filtration rate of the thickened sewage sludge did not even reach 10 mL/s. In the case of the CCS, a large dose of flocculant (≥50 mg/g) was needed to achieve higher filtration rate values (≥100 mL/s). Meanwhile, using SCF, the filtration rate reached 400 mL/s, but with an increasing dose of flocculant, i.e., higher than 6.7 mg/g, the filtration rate suddenly decreased. It indicated that the flocculating agent was overdosed, and that the dispersion was restabilized. Comparing the values of CCS and the SCF effective dose, the dose of the CCS was more than 10 times higher. However, using this flocculant, the system is not restabilized and a wide flocculation window is clearly visible.

In the flocculation process, usually only one flocculant is used, but, recently, there has been a growing interest in dual flocculant systems, in which several or more flocculation agents are used. Appositively charged polyelectrolytes, polycations, and polyanions could be added to such systems. Dual flocculation systems could prevent the overload of system (restabilization) at higher flocculant concentrations, which usually occurs when only using a single flocculant [32]. Therefore, the thickening of the surplus activated sludge by using a dual flocculant system was investigated. The dual flocculation system consisted of two appositively charged flocculants. In this case, an aqueous slurry of synthetic anionic flocculant (SAF) was added to the sludge before the flocculation with the cationic flocculant. In this way, there is a possibility to form additional electrostatic bridges between the sludge particles and increase the efficiency of the cationic starch-based flocculants by forming larger and denser flocs [33].

As shown in Figure 5a, the filtration rate of the surplus activated sludge by using CS+SAF flocculants system increased from 9 to 40 mL/s at the optimal flocculant dose of 13.5 mg/g. Hence, the SAF additive increased the flocculation efficiency of CS. On the contrary, when using a CCS+SAF dual system, the filtration rate decreased approximately 2 times at the flocculant dose of 40 mg/g (see Figure 5b).

Despite the low filtration rate of the CS + SAF thickened sewage sludge compared to that of the system containing CCS + SAF, the dual flocculant system proved to improve the flocculation efficiency of the CS. Furthermore, the turbidity of the destabilized sewage sludge filtrates was measured and the dependence of the light absorption of the obtained filtrates on the dose of the used flocculant was assessed (see Figure 6).

A significant difference was noted in the case of using CS, when the values of light absorption increased from 0.08 to 0.36 a.u. On the contrary, using CCS, the filtrate was purified well from the suspended particles. Similarly, in the studies of other authors [34], the cationic starches with high DS (0.85 and 0.97) demonstrated promising water clarification performance when treating Nile River water.

### 3.3. Dewatering of the Surplus Activated Sludge by Using Cationic Starch Flocculants

One of important prerequisites in sludge treatment is to obtain the flocs that could be readily and rapidly dewatered after the destabilization process. Therefore, the filtration efficiency of the thickened sludge was assessed using standard CST apparatus that measures the duration of the water uptake when filtering the sludge dispersion through a standard paper filter. The filtration efficiency results when dewatering the surplus activated sludge thickened using different flocculants are presented in Figure 7.

As depicted in Figure 7, the filtration efficiencies when using SCF, CS, and CCS are similar. The dewatering of the thickened surplus activated sludge using SCF, CS, and CCS were 69, 67, and 72%, respectively. Meanwhile, the optimal doses of SCF, CS, and CCS have been established to be equal to 3.1, 31.3, and 68.5 mg/g, respectively. However, an overdose of the flocculant was observed above those concentrations, which caused the filtration efficiency to decrease again. According to the literature, a flocculant dose less than 100 mg/g is considered relatively low and appropriate [4]. The optimum dose of a flocculant is one of the most important factors that affects processes in sludge dewatering. Low doses of flocculants result in weak charge neutralization and bridging effects while high doses might lead the sludge particles to be covered with flocculants (“restabilization effect”) and reduce the quality of pollutant separation [4]. This phenomenon has been also noted when analyzing the thickening of the surplus activated sludge by using SCF (see Figure 4). In contrast, our starch-based flocculant has a wide range of effective concentration resulting in protection from rapid overdosing.

Other authors [35] prepared coagulants/flocculants from acetylated corn and potato starches. The turbidity, color, pH, and electrical conductivity determined the performance of the acetylated starches as coagulants/flocculants in the treatment of industrial and urban wastewater. Acetylated starches in the concentration range from 3 to 9 g/L exhibited the best performance [35] while, in our study, starch-based flocculants demonstrated an effective flocculation process at the concentration of 0.685 g/L.

The suitability of the flocculant for sewage sludge dewatering could be also determined when assessing the formed flocs. The pictures of the dewatered raw surplus activated sludge and dewatered sludge thickened using SCF, CS, and CCS are presented in Figure 8.

The optimal flocs must be large and strong, as is obvious in the case of SCF and CCS (see Figure 8b,d). However, when using CS, the obtained flocs (see Figure 8c) were too small and rather unsuitable for effective water removal as small flocs could pass through or clog the filter, significantly reducing the filtration rate, as determined in the case of CS (see Figure 4).

Moreover, the formed flocs must not only be large but also durable and resistant to mechanical stress [36]. The strength of the flocs was assessed by measuring the filtration efficiency of the thickened sludge flocs, obtained by using an optimal dose of flocculants and exposed to mixing at different intensities before the filtration (see Figure 9).

When the mixing intensity was low (≤500 rpm), the flocs were not destroyed. However, when the mixing intensity was increased to 1200 rpm, the observed filtration efficiency decreased. Consequently, the strength of the sludge flocs obtained by using different flocculants could be ranked in the following order: CCS > CS > SCF. Moreover, it could be concluded that the flocs obtained using the CCS flocculant were most resistant to mechanical stress.

## 4. Conclusions

To obtain high-performance flocculants, high affinity for water and a sufficient availability of cationic groups of modified starches are prerequisites. Therefore, intense mechanical shearing was applied to produce an aqueous slurry of cross-linked cationic starch. In that way, the accessibility of the cationic groups was increased from 4% to 40% depending on duration of the shearing. The effectiveness of cross-linked cationic starches compared with synthetic cationic flocculants in the thickening and dewatering of municipal sewage sludge was evaluated. It was determined that the sheared cross-linked cationic starch is the most suitable for the thickening and dewatering of surplus activated sludge. Although the effective dose of this flocculant is higher when compared to that of the synthetic cationic flocculant, the cross-linked cationic starch derivatives have a number of advantages, i.e., the obtained flocs are resistant to mechanical stress; the flocculant is effective in a wide range of concentrations; and the system is not restabilized. An important advantage of the developed biobased polymer flocculant compared to existing industrial solutions is that the product is obtained as a solid material and not as a solution. This is an important issue when transporting the products from the production site to the wastewater treatment stations. Another merit is related to the use of much lower amounts of cationic reagent for the modification of starch and the use of mechanical treatment instead to obtain efficient biodegradable flocculants. However, further work involving double flocculant systems is needed to lower the dose of the biobased flocculant and to fully optimize the flocculation conditions. Moreover, the cost effectiveness, shelf life, and other important issues should be accounted for when considering the industrial production of such flocculants. To conclude, sheared cross-linked cationic starch could have a great potential as high-performance flocculant in municipal sewage sludge thickening and dewatering.

## Figures and Tables

**Figure 1 materials-16-02621-f001:**
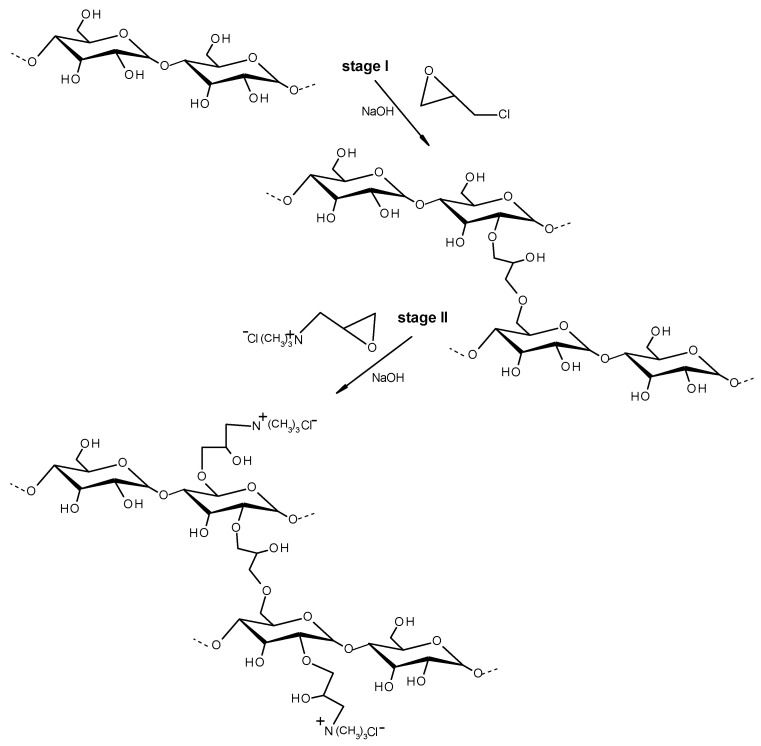
CCS synthesis scheme.

**Figure 2 materials-16-02621-f002:**
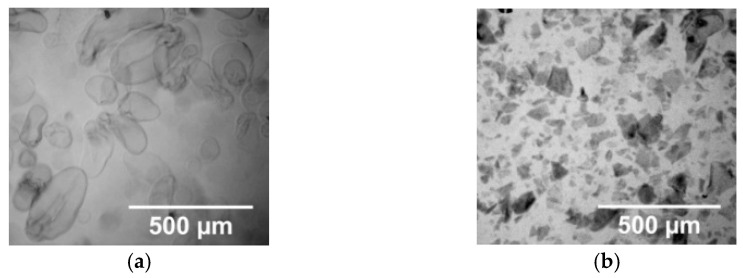
Optical microscopy images of non-sheared (**a**) and sheared CCS microgranules (**b**).

**Figure 3 materials-16-02621-f003:**
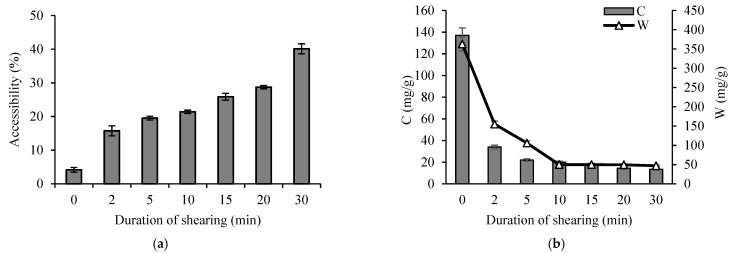
Accessibility of cationic groups (**a**) and flocculation efficiency (**b**) of CCS depending on shearing duration. C is the minimum dose of the flocculant in the presence of which the kaolin suspension destabilization occurs up to 10% of residual turbidity and W is the width of the flocculation window.

**Figure 4 materials-16-02621-f004:**
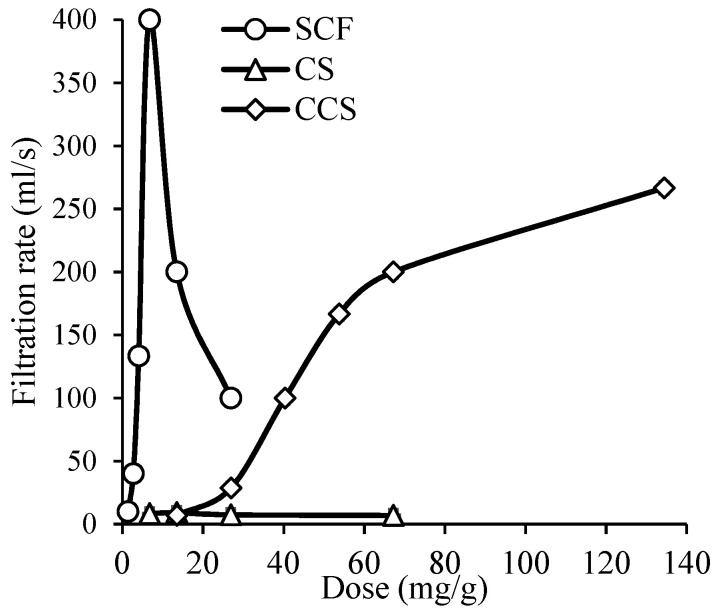
Filtration rate dependence on SCF, CS, and CCS dose used in the thickening of surplus activated sludge.

**Figure 5 materials-16-02621-f005:**
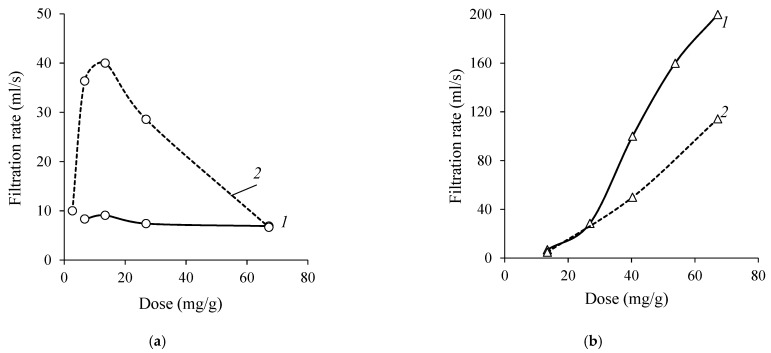
Filtration rate dependence on CS (**a**) and CCS (**b**) dose used in the thickening of surplus activated sludge: 1—sludge without SAF additive; and 2—sludge with SAF additive (0.54 mg per 1 g of sludge).

**Figure 6 materials-16-02621-f006:**
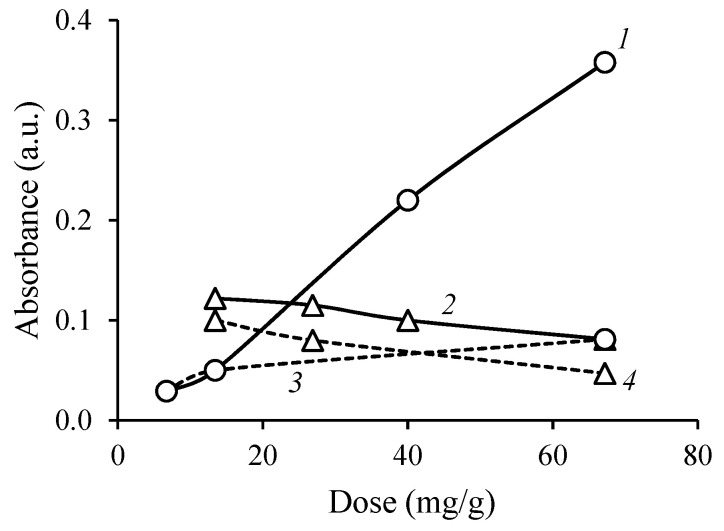
The dependence of filtrate light absorbance at 500 nm wavelength on the dose of starch derivative used in the thickening of surplus activated sludge: CS (1), CCS (2), CS + SAF (3), and CCS + SAF (4).

**Figure 7 materials-16-02621-f007:**
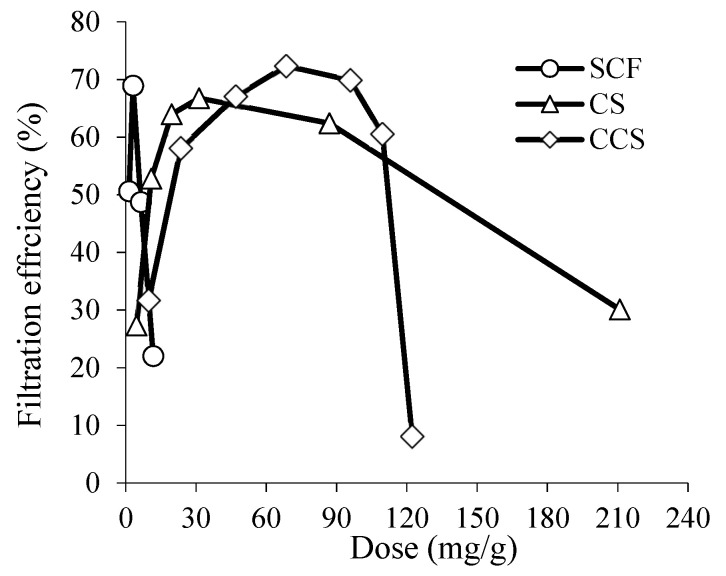
The dependence of filtration efficiency on SCF, CS, and CCS doses when dewatering thickened surplus activated sludge.

**Figure 8 materials-16-02621-f008:**
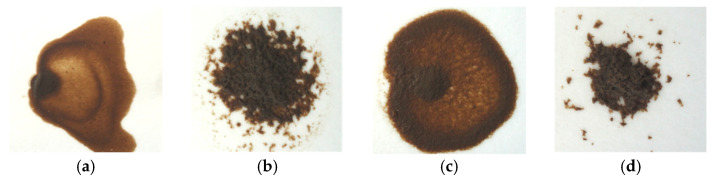
Photographs of dewatered surplus activated sludge flocs: raw sludge (**a**), thickened using SCF (**b**), thickened using CS (**c**), and thickened using CCS (**d**).

**Figure 9 materials-16-02621-f009:**
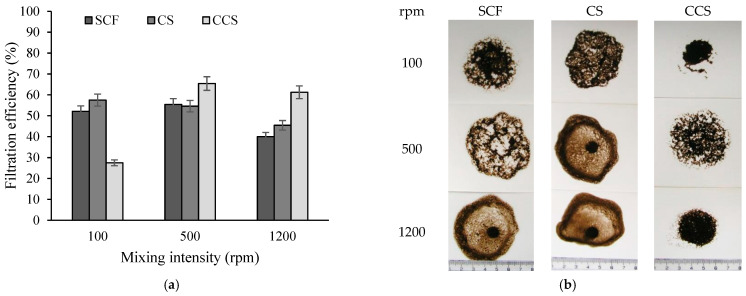
(**a**)—The dependence of filtration efficiency of surplus activated sludge thickened using SCF, CS, and CCS on the mixing intensity of sludge dispersion; and (**b**)—photographs of dewatered surplus activated sludge flocs when thickened using SCF, CS, and CCS and exposed to the mixing at different intensities.

**Table 1 materials-16-02621-t001:** A comparison of various coagulants and flocculants [4].

Coagulants/ Flocculants	Agent	Dose (mg/g Dry Solid)	pH Range	Dewatering Efficiency	Cost	Toxicity
Inorganic coagulant	Polyaluminium chloride, FeCl_3_	High (>100)	Moderate (5.0–9.0)	Moderate	Low	High
Synthetic polymeric flocculants	Cationic polyacrylamide, Poly(acrylic acid)	Low (<100)	Wide (2.0–12.0)	Good	High	High
Natural polymeric flocculants	Starch, chitosan	Low (<100)	Wide (2.0–12.0)	Good	Moderate	Low

## Data Availability

Not applicable.

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
