# Peer review of "Production of Cationic Starch-Based Flocculants and Their Application in Thickening and Dewatering of the Municipal Sewage Sludge"

_materials, 2023, doi:10.3390/ma16072621_

Round 1

Reviewer 1 Report

The article concerns modified starch flocculants. This is a very interesting article on cationic starch flocculants and cross-linked cationic starch flocculants in thickening and dewatering of surplus activated sewage sludge. The starch materials were compared with a synthetic cationic flocculant.

The article is well written but needs minor corrections and additions:

1. The literature should be complemented and extended with more recent items.

2. There is no discussion regarding the comparison of the results for the obtained materials with other similar ones. I suggest expanding the article with the mentioned discussion.

3. Potato starch requires more precise characteristics, e.g. average molecular weight or amylose and amylopectin content

Reviewer 2 Report

Title: Production of cationic starch-based flocculants and their application in thickening and dewatering of the municipal sewagesludge

Recommendation: Minor revisions needed as noted.

Every interesting paper but there is few questionable points that the authors should try to more clearly address.

1. There are similar article published in the literature? What is very unique in this work?

2. The novelty of the current work must be explained clearly in the introduction section. The research filled is not fully explained. Explain it in detail at end of the introduction section.

3. What were the main challenges you faced during your study, and how did you overcome them?

4. What are the optimal conditions for using the flocculant in thickening and dewatering of municipal sewage sludge?

5. How does the flocculant compare to other commonly used flocculants in terms of its effectiveness and efficiency?

6. What are the environmental implications of using the cationic starch-based flocculant in wastewater treatment?

7. Are there any potential health risks associated with the use of this flocculant, either during its production or application?

8. What are the limitations of the study and what further research is needed in this area?

9. What are the potential economic benefits of using this flocculant compared to other flocculants currently in use?

10. How can the findings of this research be applied in industrial wastewater treatment processes?

11. How does your study contribute to the existing literature on the topic?

12. Between the digit and the Unit, a space should be added.

13. Use only the same form of the units, i.e min or minutes, in the whole manuscript. 

Reviewer 3 Report

In the presented draft, the authors described their works synthesizing cross-linked cationic starch-based flocculants (CCS) and their application in thickening and dewatering municipal sewage sludge (MSS). The topic is exciting and of significant positive environmental impact. However, some disadvantages severely weaken this draft. Here are the comments.

1.     Abstract. “Modified starch flocculants were obtained by using chemical modification and mechanical shearing of potato starch microgranules.” can be rephrased into “Potato starch microgranules were chemically modified and mechanically sheared to produce modified starch flocculants.”

2.     The authors failed to address why we need CCS in sludge thickening and dewatering. Compared to the 69% filtration efficiency from the synthetic cationic flocculant (SCF) and the 72% CCS. No great improvement was shown. Did you produce CCS or CS cheaper than SCF? No economic analysis was provided. Also, the dosage of the newly synthesized flocculants is greater. Therefore, I failed to see the novelty of this work.

3.     No detailed discussion. What the authors provided is just like an experiment report. There is no detailed discussion for the synthesized materials and what could lead to the better dewatering ability (which you didn’t have).

4.     Meaningless results. Suppose your new flocculants need much more dosage but cannot significantly promote dewatering efficiency. Why do we need it?

5.     Relatively old reference. Only 2/22 of the reference papers were published in the past five years (2019-2023). The authors should review more recent works for dewatering sewage sludge and utilize sludge. The authors can cite the following papers: Chemical Engineering Journal, 349, 737-747.; Separation and Purification Technology, 306, 122620.; Journal of Water Process Engineering, 46, 102567. Science of The Total Environment, 153328.; RSC advances, 12(31), 20379-20386.

6.     Too many self-citations. References 11, 16, 17, 18, and 19 are all from the draft authors. Considering you only cited 22 papers, that’s a lot. 
